# Corpus Statistics Empowered Document Classification

**Farid Uddin** [1] , **Yibo Chen** [2], **Zuping Zhang** [1,*] and **Xin Huang** [2]

1    School of Computer Science and Engineering, Central South University, Changsha 410083, China;
     faridbabu@outlook.com
2    Information and Communication Branch, State Grid Hunan Electric Power Company Limited,
     Changsha 410021, China; chenyibo8224@163.com (Y.C.); huangx61@126.com (X.H.)
*    Correspondence: zpzhang@csu.edu.cn

**Abstract:** In natural language processing (NLP), document classification is an important task that relies on the proper thematic representation of the documents. Gaussian mixture-based clustering is widespread for capturing rich thematic semantics but ignores emphasizing potential terms in the corpus. Moreover, the soft clustering approach causes long-tail noise by putting every word into every cluster, which affects the natural thematic representation of documents and their proper classification. It is more challenging to capture semantic insights when dealing with short-length documents where word co-occurrence information is limited. In this context, for long texts, we proposed *Weighted Sparse Document Vector* (WSDV), which performs clustering on the weighted data that emphasizes vital terms and moderates the soft clustering by removing outliers from the converged clusters. Besides the removal of outliers, WSDV utilizes corpus statistics in different steps for the vectorial representation of the document. For short texts, we proposed *Weighted Compact Document Vector* (WCDV), which captures better semantic insights in building document vectors by emphasizing potential terms and capturing uncertainty information while measuring the affinity between distributions of words. Using available corpus statistics, WCDV sufficiently handles the data sparsity of short texts without depending on external knowledge sources. To evaluate the proposed models, we performed a multiclass document classification using standard performance measures (precision, recall, f1-score, and accuracy) on three long- and two short-text benchmark datasets that outperform some state-of-the-art models. The experimental results demonstrate that in the long-text classification, WSDV reached 97.83% accuracy on the AgNews dataset, 86.05% accuracy on the 20Newsgroup dataset, and 98.67% accuracy on the R8 dataset. In the short-text classification, WCDV reached 72.7% accuracy on the SearchSnippets dataset and 89.4% accuracy on the Twitter dataset.

**Keywords:** classification; text mining; natural language processing; machine learning; corpus statistics

## 1. Introduction

Language modeling with binary one-hot word encoding is higher dimensional and sparse with no semantic information. As a result, the word analogy is missing; e.g., the distance between word vectors represents only the difference in alphabetic ordering. However, point vector representation of words in the embedding space like word2vec [1] and GloVe [2,3] contain semantic information. Representing a dense low-dimensional fixed-length document vector is much more expensive and complicated. Moreover, it is challenging to infer unseen documents during the test process. The simplest method to get document embedding is the weighted averaging of word embeddings in the document [4]. Document Vector through Corruption (Doc2VecC) [5] is a significant study that shows how a simple weighted averaging technique combined with a simple noise-eliminating procedure can be effective. However, Doc2VecC did not cover the underlying themes of the document. Sparse Composite Document Vector (SCDV) [6] addresses this limitation by using word embeddings and the Gaussian mixture clustering to generate the document vector, which also overcomes the shortcomings of the hard-clustering

approach [7]. SCDV shows significant improvement in the downstream natural language processing (NLP) tasks, including document classification. However, SCDV inherits noisy tails from the Gaussian mixture clustering that is not appropriate for the document containing multiple sentences [8]. Another vital issue ignored by most document representation methods is ignoring potential terms in the corpus, which is essential for understanding deep semantic insight of the documents.

It is hard to encode the richness of semantic insights for short-length documents where word co-occurrence information is limited [9]. Therefore, many works suggest importing semantic information from external sources [10–12]. However, accessing information from external sources (e.g., Wikipedia) may cause irrelevant noise corresponding to the current short text corpus.

In this context, we aim to develop corpus statistics-based semantically enriched vectorial representation of the noisy long and sparse short texts for the multi-class document classification performance with the objectives of solving the following research questions:

- How to efficiently model noisy long and sparse short texts for the classification performance?
- How to efficiently encode the semantically enriched vectorial representation of documents using available corpus statistics to enhance classification performance?
- Is it possible to model efficient sparse short-length documents utilizing available corpus statistics instead of depending on external knowledge sources?
- How can we utilize potential words in the document for deep thematic insights?

Sparse Composite Document Vector with Multi-Sense Embeddings (SCDV-MS) [13] forces discarding the outliers from the clustering output to eliminate long-tail noises in the SCDV, which applies a hard threshold that may hinder the thematic representation of documents. Moreover, representing proper expressive documents depends upon modeling the underlying semantic topics in the correct form [14], which requires capturing deep semantics insights buried in words, expressions, and string patterns [15]. Hence, for the noisy long texts, we proposed *Weighted Sparse Document Vector* (WSDV) that embodies important words emphasizing capability using *Pomegranate General Mixture* model [16], and a soft threshold-based noise reduction technique.

It is challenging to capture semantics insights in document modeling with sparse short texts. The probability distribution of words captures better semantics than the point embedding approach (e.g., word2vec) [17] as it generalizes deterministic point embeddings of the terms using the mean vector, and the covariance matrix holds uncertainty of the estimations. Hence, instead of depending on external knowledge sources, we proposed corpus statistics empowered *Weighted Compact Document Vector* (WCDV), which emphasizes potential terms while performing probability word distribution using the weighted energy function. In WCDV, we employ the *Multimodal word Distributions* [18] that learns distributions of words using the Expected Likelihood Kernel [19], which computes the inner product between distributions of words to get the affinity of word pairs. However, every word in a document does not hold the same importance; some are used more frequently than others, indicating their importance in the corpus. It is required to emphasize the frequently used words, especially when word co-occurrence information is limited (e.g., microblogging, product review, etc.). Therefore, to preserve the word frequency importance, we proposed *Weight attained Expected Likelihood Kernel* which considers term frequency-based point weights while measuring the partial log energy between distributions in the *Multimodal word Distributions* [18].

The organization of the remaining parts of the paper is as follows. Section 2 contains discussions about related works. Section 3 explains the methodologies used in the proposed approaches. Section 4 represents the functionality of the proposed approaches. Section 5 carries out the analysis of the experiments and obtained results. Finally, Section 6 concludes the article.

## 2. Literature Review

Word embeddings models ignore side information (e.g., document labels) while learning embeddings from enormous document corpora. To improve word representation and text classification accuracy, Linear, Y. et al. [20] proposed to use document labels as the global context both in the local neural network model and the global matrix factorization framework. Obayes, H.K. et al. [21] combined GloVe and bidirectional long short-term memory (BiLSTM) recurrent neural network for better sentiment classification, which causes expensive computation and no guidance for documents containing multiple sentences. Yang, Z. et al. [22] proposed Hierarchical attention networks (HAN) for document classification, which maintain a hierarchical structure of word to sentence (building sentence from words) and sentence to document (aggregating sentences to a document representation). Zhang, Z. et al. [23] proved that the TFIDF algorithm with the combination of Naive Bayes has significance in the text classification task compared to many complex models.

Recently, transformers-based models [24,25] became more prevalent in downstream Natural Language Processing (NLP) tasks (e.g., document classification). Wang, B. and-linebreak Kuo, C-C.J. [26] proposed SBERTWK for sentence embedding, which trains on both word and sentence level objectives but no guidance for representing a document that contains multiple sentences. However, the transformer-based model requires enormous computational resources. Sanh, V. et al. [27] introduced a distilled version of BERT called DistilBERT, which is smaller, faster, cheaper, and lighter than other transformers-based models.

Mapping sentences to a fixed-length embedding vector using Universal Sentence Encoder (USE) based method [28] also got success in the downstream Natural Language Processing (NLP) task. The sentence analysis method made by combining Universal Language Model Fine-tuning (ULMFiT) with the Support Vector Machine (SVM) [29] is capable of performing document classification using a small amount of data but has higher computational complexity.

Yet, K.S. et al. [30] proposed document embedding along with their uncertainty called the Bayesian subspace multinomial model (Bayesian SMM) to capture better semantics. It is a generative log-linear model that learns to represent documents in the form of the Gaussian distributions and encodes uncertainty in the covariance matrix but holds only a single mode of words. Therefore, encoded uncertainty might diffuse spontaneously; the mean vector can be pulled in one direction and represents one particular meaning by leaving others not representing [31]. Different senses of a word lie in the linear superposition of standard word embeddings [32] and the Gaussian mixture model holds multiple modes to represent distinct meanings of words.

For the long texts classification, we proposed WCDV, which represents documents with uncertainty estimations in the distribution of words using Gaussian Mixtures distributions for short-length document classification. We proposed WSDV using the Pomegranate General Mixture model for the long texts classification. Both WSDV and WCDV accommodate polysemous terms and train on the labeled documents corpus for better classification performance.

Noisy topics are outliers prone, thus less coherent and less expressive. Newman, D. [33] regularized the LDA-based topic model where only the higher frequency terms allow into the word dependencies sparse covariance matrix. This model executes two prime steps. Firstly, measuring the point weight of each word in the vocabulary, and secondly, putting a threshold point to eliminate lower weighted words from the covariance matrix. Mittal, M. et al. [34] introduced automated K-means clustering, where they applied a threshold point to decide whether or not to create a new cluster for the objects. This approach prohibits outlier tendency by accommodating lower probability objects into a new cluster. Gupta, V. et al. [13] introduced SCDV-MS, which removes noise by applying a hard threshold on the fuzzy word cluster assignments, which proved better classification performance and lower space and time complexity than SCDV [6].

In contrast, the proposed WSDV contains more natural noise removal techniques using a soft threshold and more efficient sparse vectorial representation for the long text (e.g., removing first principle components).

To capture better corpus semantics, Sia, S. et al. [35] introduced weighted data clustering on pre-trained word embeddings, where they also proved the effectiveness of re-ranking the top words in a cluster for better representative topics. Similarly, Gebru, I.D. et al. [36] proposed a Gaussian mixture-based weighted data clustering method called WD-GMM that demonstrates how the point weight of datum affects the covariance matrix and leads to better clustering. Inspired by them, we proposed WSDV, which extends the clustering process on the weighted data for the multi-class document classification performance.

Short texts are sparse due to limited word co-occurrence, which requires special treatment to capture hidden semantic information [37,38]. Pretrained word embedding over large external corpora is a common remedy for dealing short length documents. Zuo, Y. et al. [39] proposed a word embedding-enhanced Pseudo-document-based Topic Model (WE-PTM) to leverage pre-trained word embeddings that is essential for alleviating data sparsity. Instead of incorporating external knowledge sources, Zhang, P. and He, Z. [40] proposed an ensemble approach by exploiting both word embeddings and latent topics in sentence-level sentiment analysis for sentence polarity detection.

Therefore, for semantically enriched short-length document representation, instead of importing information from external knowledge sources, we employ *Multimodal word Distributions* [18] to capture uncertainty in the distribution of word embeddings for the vectorial representation of documents.

The contextual analysis-based model emphasizes potential terms that capture better semantics insights and boost classification performance [41]. Xu, J. et al. [42] proposed a convolutional neural network-based model, which incorporates context-relevant concepts into text representation for uplifting short text classification performance, but it requires expensive computational capacity.

In WCDV, we use the weighted energy function to emphasize potential terms in the short texts corpus.

Weighted Kernel Density Estimation (WKDE) [43,44] based on point weights has proved effective. For the semantic similarity measuring task, constant weighting assumption-based semantic similarity [45] measure between two concepts/words holds better performance for the semantic representation of the concept/words but holds the same weighting relevance. Later, it found that the weight propagation mechanism [46,47] for augmenting input with semantic information achieves desired performance and removes the same weighting curse for concepts/words. Recently, Liu J. et el. [48] introduced a weighted kernel mechanism for the weighted k-means multi-view clustering, where they redefined the objective by assigning weights to the cluster level instead of global weighting for each view and outperforming the existing objective.

Inspired by their work, we proposed a novel word frequency concerned energy function called *Weight attained Expected Likelihood Kernel* for computing affinity between word pairs. To capture better segments in the WCDV, we modify the objective of the *Multimodal word Distributions* [18] by applying the newly proposed energy function and employ the modified *Multimodal word Distributions* for the topic distribution of words in WCDV.

## 3. Methodology

Using available corpus statistics leads to success in many downstream Natural Language Processing (NLP) tasks. Semantic information of documents plays a potential role in their classification [49]. Topic modeling is the standard approach for unveiling the underlying semantics of documents [50]. Word embeddings and clustering are the best partners as clustering methods can utilize available corpus statistics for exploratory document analysis [51]. We use weighted data clustering to obtain the underlying topis and represent document vectors utilizing corpus statistics in different stages.

### 3.1. Clustering on Weighted Data

One of the most optimized computational complexity topic modelings is the Latent Dirichlet Allocation (LDA) [52], which computes the probability of a word being in a particular topic based on the statistics of how many times this word appears in this topic. Inspired by LDA, Sia, S. et al. [35] studied various techniques to define weight for each data point and found that the *term frequency* (tf) performs better. Inspired by their work, we use the *term frequency* (tf) to form the point weight for each data point in the training corpus.

$$term\ frequency,\ tf = \frac{n_t}{\sum_{t'} n_{t'}}, \tag{1}$$

$$point\ weights,\ \omega = tf + 0.01. \tag{2}$$

Here, $n_t$ is the count of word type $t$, and $\sum_{t'} n_{t'}$ is the total count of all word types in the corpus. The numerical constant 0.01 manipulates *tf*, such that $\sum \omega \neq 1$.

Let $x \in R^d$ be the random vector following a multivariate Gaussian distribution with the mean $\mu \in R^d$ and the covariance $\Sigma \in R^{d \times d}$, namely $p(x|\theta) = \mathcal{N}(x; \mu, \Sigma)$ with the notation $\theta = (\mu, \Sigma)$. Let $\omega > 0$ be a weight indicating the relevance of the observation $x$ such that the higher the weight, the stronger the impact of $x$, that is observing $x\ \omega$ times when incorporating $\omega$ as the weight relevance of the data point $x$. The likelihood function is thus $\mathcal{N}(x; \mu, \Sigma)^\omega$. The power $\omega$ is not a probability distribution as it does ont sum to 1 (one); it plays the role of the precision [36] and is different for each datum $x$. So,

$$p(x, \theta, \omega) = \mathcal{N}(x; \mu, \Sigma)^\omega = \mathcal{N}(x; \mu, \frac{1}{\omega}\Sigma) . \tag{3}$$

From Equation (3), a Gaussian mixture based density function to represent the probability distribution of a word (data point) $x$ with $k$ components then derives as bellow:

$$\begin{aligned}
f(\vec{x}) &= \sum_{i=1}^{k} p_{x,i}\ \mathcal{N}(x; \mu, \frac{1}{\omega}\Sigma_{x,i}) \\
&= \sum_{i=1}^{k} \frac{p_{x,i}}{\sqrt{2\pi|\frac{1}{\omega}\Sigma_{x,i}|}} e^{-\frac{1}{2}(\vec{x} - \mu_{x,i})^T \Sigma_{i=1}^{-1}(\vec{x} - \mu_{x,i})}
\end{aligned} \tag{4}$$

where, $p_{x,i}$'s are mixture coefficients satisfy $p_{x,i} > 0$ and $\sum_{i=1}^{k} p_{x,i} = 1$. $\vec{x}$ is the vector representations of word $x$, and $\omega > 0$ contains weights associate with each data point $x$.

Gaussian Mixtures Distributions and its automatic variant [53] do not allow weights as a parameter for data clustering. As a result, SCDV and SCDV-MS do not support clustering on the weighted data. An alternative implementation to the Gaussian Mixtures Distributions called *Pomegranate General Mixture* model [16] allows the mixtures of arbitrary distributions with the same dimensionality components. The *Pomegranate General Mixture* model allows weights parameter along with the input data for the soft clustering that estimates a covariance matrix for each component. The weights parameter contains the initial weight for each data point; if the weights parameter does not utilize, then it initializes the same weight for all data points.

### 3.2. Moderate Clustering

The Gaussian mixture clustering is outlier-prone as it poses the soft clustering concept. Some alternative models to handle outliers are the mixture of t-distributions [54], and its variants [55–60]. The soft clustering causes data outliers in SCDV by allowing the lower probability words in the clusters. An outlier-prone data point is far from the cluster center with a low probability score. SCDV-MS shows removing a particular portion of the data points with low probability scores alleviate noise, which puts a hard threshold. To reduce outliers in WSDV, we introduce a dynamic soft threshold constraint to moderate the soft

clustering approach called *moderate clustering*. This constraint makes sure only words $w$ out of W (where $w < W$) with a higher probability score can be part of a cluster [32,61]. In WSDV, after the convergence of the clustering process, we calculate a threshold point using Equation (5). We put 0 (zero) to the values below the threshold point in the word probability matrix, which removes the words from the clusters by not allowing every word but those above the threshold point.

Let $\rho$ be the threshold point, $\gamma$ be the percentage probability threshold, and $\phi_k$ be the probability of a word being in the $k$th cluster. So,

$$\phi_k = \begin{cases} \phi_k & \text{if } |\phi_k| \geq \rho \\ 0 & otherwise \end{cases} \tag{5}$$

where, $\rho = \frac{\gamma}{100} * t$; $t = \frac{|\phi_{k_{min}}| + |\phi_{k_{max}}|}{2}$.

The value of the percentage probability threshold $\gamma$ needs to be adjusted, we found that for $k = 20$ in 20Newsgroup, $k = 8$ in R8 and $k = 7$ in AgNews, $\gamma = 7$ works better. The moderate clustering strategy divides the embedding space into the number of clusters semantic subspaces [8]. As a result, WSDV is more expressive for document representation than SCDV.

### 3.3. Re-Ranking the Top Words

In distribution-based clustering algorithms like the Gaussian Mixture model, the top words are those that have the highest probability in the cluster as a result closest to the cluster center $c^{(k)}$ (center of the $k$th cluster) that is for the top $J$ words

$$argmin_{J:|J|=10} \sum_{j \in J} f(x_j) | c^{(k)}, \Sigma_k .$$

The top $J$ words of a cluster may represent a sensitive topic without the guarantee of having important words closest to the cluster center [35]. A simple re-ranking of the terms using corpus statistics gives a better topical representation of the clusters. The re-ranking has an extra $O(n \log(n_k))$ cost factor for the $n$ unique terms in the vocabulary, where $n_k$ is the average number of elements in the $k$thcluster. Differ from SCDV, after the convergence of the weighted data clustering, we re-rank the top words using simple *word frequency* of the corpus that leads to relevant top $J$ words closest to the cluster center as below:

$$argmin_{J:|J|=10} \sum_{j \in J} f(x_j) | c^{(k)}, \frac{1}{\omega} \Sigma_k$$

where weight vector $\omega$ contains different *point weight* for each term $x$ in the corpus.

### 3.4. Kernel Based Similarity

Like any other input objects, word vectors can be implicitly mapped into the Hilbert space with a base kernel and fit a Gaussian distribution on them [62]. This process combines the advantage of discriminative learning algorithms and kernel machines with generative modeling. The kernel evaluates by integrating the product of the fitted generative models on the corresponding data points [63].

Let $K$ is the kernel, and two input objects $x$ and $x'$ are mapped over the Hilbert space $\langle \rangle_{L_2}$ using a mapping function $\Phi$

$$K(x, x') = \langle \Phi(x), \Phi(x') \rangle_{L_2} \tag{6}$$

Instead of direct operation on the training and the testing objects, kernel-based algorithms focus on evaluating the value of the kernel function for each pair of objects [64]. Objects $x_i \epsilon \mathbf{R}^n$ generally represented as vectors, kernel $K$ is a closed-form positive definite function on $\mathbf{R}^n$ such as Gaussian RBF

$$K(x, x') = e^{\frac{-\| x - x' \|^2}{(2\sigma^2)}} \tag{7}$$

The input space literally can be anything as long as the applied function on it is positive definite and holds reliable similarity measure between examples, which satisfies the condition below:

$$\sum_{i,j=1}^{m} C_i C_j \, K(x_i, \, x_j) \; \geq \; 0$$

It ensures that the Kernel $K$ is a positive (semi-) definite symmetric similarity measure for any $m \epsilon \mathbf{N}$ and set of examples $x_1, x_2, x_3, \, \dots \, , x_m \, \epsilon X$ for representing input space with coefficients $C_1, C_2, C_3, \, \dots \, , C_m$. This conditioning results in a massive expansion of novel kernels in different fields, such as kernels on the statistical manifold [65], Gapped String Kernels [66] and so on. There are several kernel-based similarity or distance measures for distributions, e.g., Semantic kernel-based Neural embeddings [67], Fisher kernel-based Bayesian network [68], Heat kernel-based Manifold learning [69], Kullback-Leibler (KL) divergence for Die cracks detection [70] and so on. The KL-divergence is not straightforward as kernels and is not positive definite. The disadvantage of this kind of asymmetric local approximation model is that the exponential family only generates a linear Fisher kernel. The Heat kernel is an alternative nonlinear kernel capable of dealing with the statistical manifold. However, it is still not fully capable of handling the exponential family or the mixture model in the closed-form.

From Equation (6) kernel between probability distributions, $p$ and $p^{'}$ on space $x$ can draw as

$$K(x, x^{'}) \; = \; K(p, p^{'}) \; = \; \langle p(x), p^{'}(x) \rangle_{L_2} \tag{8}$$

The probability product kernel between distrains $p$ and $p^{'}$ is

$$K(x, x^{'}) \; = \; \int_x p(x)^{\rho}, p^{'}(x)^{\rho} \, dx \; = \; \langle p, p^{'} \rangle_{L_2} \tag{9}$$

where $\rho$ is a positive constant, such that $p^{\rho}, p^{'\rho} \, \epsilon \, L_2(x)$; $L_2(x)$ represents the Hilbert space. Different values of $\rho$ has significance in kernel's evaluation, e.g., $\rho = 1/2$ involve statistical affinity between distributions

$$K(p, p^{'}) \; = \; \sqrt{p(x)} \sqrt{p^{'}(x)} \, dx \tag{10}$$

Equation (10) is known as the Bhattacharyya kernel [19]. It is commonly known as Bhattacharyya affinity between distributions in the statistical literature, which is related to the symmetric approximation of the Kullback-Leibler (KL) divergence known as the Hellinger's distance

$$H(p, p^{'}) \; = \; \frac{1}{2} \int (\sqrt{p(x)} \sqrt{p^{'}(x)})^2 \, dx \tag{11}$$

by $H = \sqrt{2 - 2k}$. The Bhattacharyya kernel has a special property $k(p, p^{'}) = 1$. For $\rho = 1$, the kernel has another special form that behaves as the expectation of one distribution under the other:

$$\begin{aligned} K(p, p^{'}) &= \int_x p(x)^{\rho}, p^{'}(x)^{\rho} \, dx \\ &= E_p[p^{'}(x)] = E_{p^{'}}[p(x)] \end{aligned} \tag{12}$$

Equation (12) is known as the Expected Likelihood Kernel, which is easy to evaluate through the sampling methods when the closed-form does not exist.

## 4. Proposed Approaches

As stated earlier, we proposed two document representation methods derived from corpus statistics for classification tasks. Clustering regular documents (long texts) cause irrelevant noise. To capture semantically enriched topics and redeem noisy long tail, we proposed *Weighted Sparse Document Vector* (WSDV).

It is always challenging to capture hidden semantics when word co-occurrence is limited. To overcome the data sparsity challenge in capturing hidden semantics for short texts, we proposed *Weighted Compact Document Vector* (WCDV), which captures the uncertainty of the topic distribution of words using a novel energy function.

### 4.1. Weighted Sparse Document Vector (WSDV)

Proposed *Weighted Sparse Document Vector* (WSDV) is closely related to two recent sparse documents representing methods, namely SCDV: *Sparse Composite Document Vectors using soft clustering over distributional representations* [6] and *Improving Document Classification with Multi-Sense* (SCDV-MS) [13]. However, WSDV poses some functionality differences that made WSDV robust under a small number of topical settings over the other two.

In contrast to SCDV and SCDV-MS, WSDV performs clustering on the weighted data to capture the potential terms using *Pomegranate General Mixture* model, which plays a vital role in topics generation and document embeddings. WSDV and SCDV-MS perform a data sparsity mechanism on the converged clusters to remove the noisy tail, whereas SCDV does not remove noise at the cluster level. Instead of applying the hard threshold as in SCDV-MS, WSDV calculates the sparsity threshold based on the word-cluster probability described in Section 3.2. All three methods (SCDV, SCDV-MS, and WSDV) use *inverse document frequency* (idf) weighting in the topic distribution of words while capturing the topic proportion of the document. However, unlike SCDV and SCDV-MS, WSDV uses *smooth inverse frequency* (sif) [71] to capture better semantics while creating document vectors.

SCDV uses a sparsity threshold on the document embeddings to reduce noise for getting final document vectors. In contrast, SCDV-MS does not remove noise at the document embeddings level but removes outliers from the clusters. However, WSDV removes clustering outliers and the first principle components from the document embeddings and finally makes document vectors sparse by applying a sparsity threshold to ensure high-quality document vectors.

For the weighted data clustering using Algorithm 1 on the multisense text corpus, WSDV follows similar steps as SCDV-MS to get a multisense text corpus.

Figure 1 represents the functionality of getting the underlying themes of the documents by moderating the soft clustering. Figure 2 represents the overall functionality of the proposed *Weighted Sparse Document Vector* (WSDV).

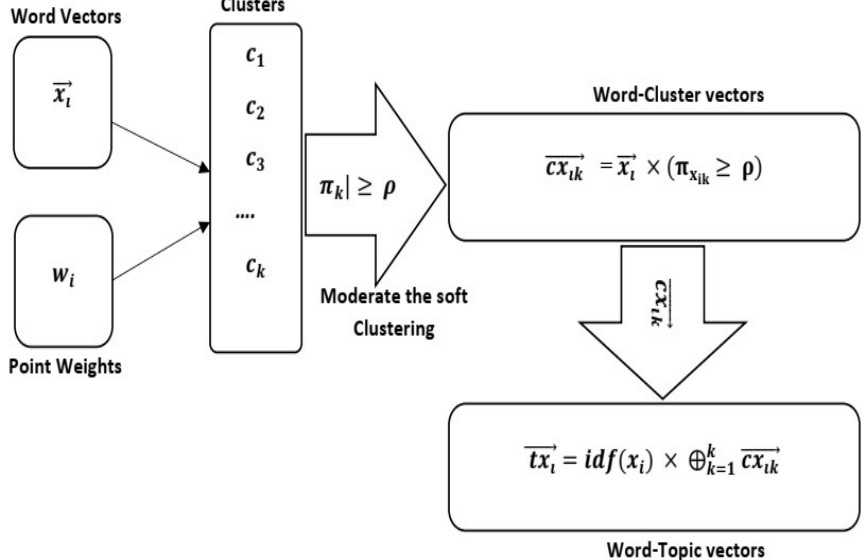

**Figure 1.** Topics formation using moderate clustering and corpus statistics.

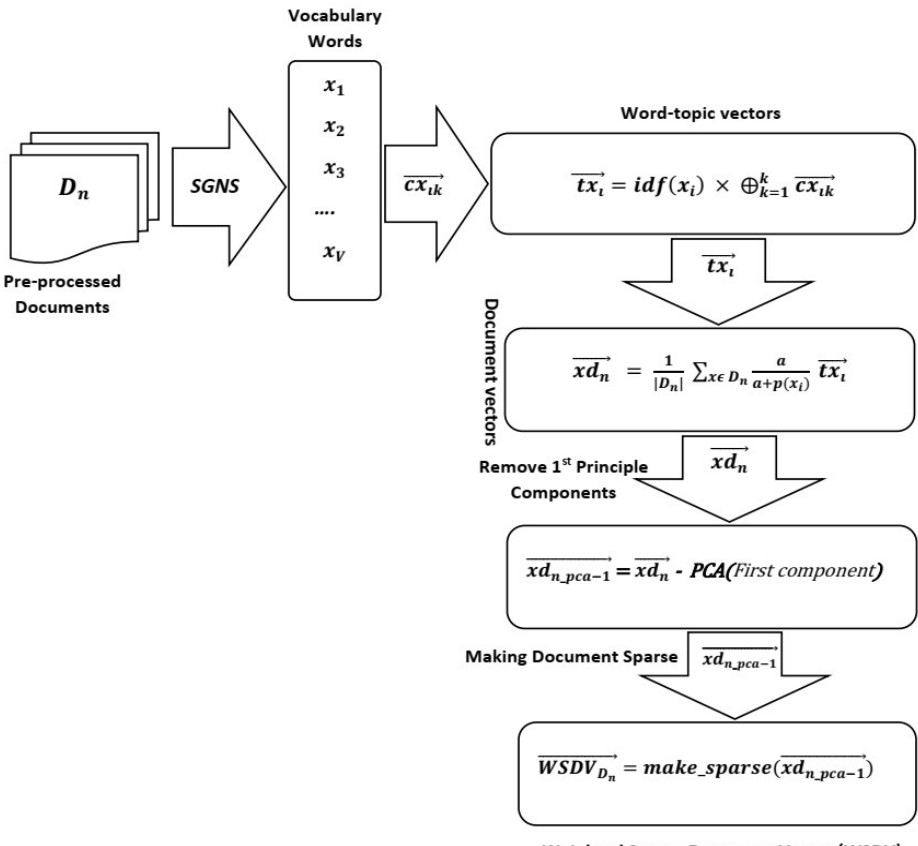

**Figure 2.** Weighted Sparse Document Vector formation in to $d \times k$ dimensional embedding space.

WSDV learns a $d$ dimensional embeddings for every word in the vocabulary $V$ using SkipGram negative sampling (SGNS) [1].

In this process, WSDV calculates the weight of each unique term using Equations (1) and (2). Unlike SCDV, besides the input text, WSDV takes this *point weights* associated with each data point as the input. After the convergence of the Pomegranate General mixture model, it applies the probability threshold (soft threshold) constraint to alleviate outliers by moderating the soft clustering outcomes using Equation (5). In the document representation process, for each word, WSDV creates $k$ different word-cluster vectors $\vec{cx}_{ik}$ of $d$ (Figure 1) dimensions by weighting the word vector with its probability score in the particular clusters $\vec{cx}_{ik} = \vec{x}_i \times (\pi_{x_{ik}} \geq \rho)$, where the term $(\pi_{x_{ik}} \geq \rho)$ makes sure only the words with a probability equal or above the threshold point $\rho$ belong to a particular cluster. Next, it weights words $x_i$ with the *inverse document frequency* (idf) and concatenate all words in each $k$ word-cluster vectors $\vec{cx}_{ik}$ to construct word-topic vectors $\vec{tx}_i = idf(x_i) \times \oplus_{k=1}^{K} \vec{cx}_{ik}$ similar to SCDV.

Applying the moderate clustering approach, embedding space is now partitioned into $k$ different semantic sub spaces for each document in $D_n$. Unlike SCDV, instead of direct sum of word–topic vectors $\vec{tx}_i$ we take weighted average of each word for each word-topic vector in a document, weighted by the *smooth inverse frequency* (sif) [71] to form the document vector $\vec{xd}_n = \frac{1}{|d_n|} \sum_{x \in d_n} \frac{a}{a+p(x_i)} \vec{tx}_i$. Document vector $\vec{xd}_n$ is represented now into $d \times k$ dimensional embedding space. As discriminative document representation ensures better classification results [72], we remove the first principle component (common context) to eliminate noise and redundancy from the document vector [8]. Finally, make the document vector sparse by following the same procedure as SCDV. Figures 1 and 2 depict the process described for WSDV.

From the discussion above, we derive Algorithm 1 for the proposed Weighted Sparse Document Vector (WSDV). In Algorithm 1, lines 1 and 2 are the initial procedure to represent

a document. Lines 3 to 9 are responsible for the topic construction, where line 6 moderates the soft clustering to construct word-cluster vectors $\vec{cx}_{ik}$, and line 8 constructs word-topic vectors $\vec{tx}_i$. Lines 10 to 17 are responsible for the sparse document representation, where line 13 constructs the document vector $\vec{xd}_n$ by computing the *smooth inverse frequency* (sif) [71] weighted average of topic-specific words. Line 15 removes the first principle component from each document vector $\vec{xd}_n$, and line 16 makes the document vector sparse.

---

**Algorithm 1:** WSDV

**Input:** Documents $D_n$; $n = 1 \dots N$
**Output:** Document Vectors $W\vec{SD}V_{D_n}$, in to $d \times k$ embedding space; $n = 1 \dots N$

1　Getting word embedding $\vec{x}_i$ of $d$ dimensions for each word $x$ using word2vec skip-gram with negative sampling(SGNS)
2　Calculate *idf*, *tf* and the *point-weights* $\omega$ for the corpus
3　Cluster word vectors $\vec{x}_i$ in to $K$ clusters considering the initial *point weights* $\omega$ using *Pomegranate General Mixture* model
4　**for** *each word $x_i$ in the vocabulary* **do**
5　　**for** *each cluster $c_k$* **do**
6　　　$\vec{cx}_{ik} = \vec{x}_i \times (\pi_{x_ik} \geq \rho)$
7　　**end**
8　　$\vec{tx}_i = idf(x_i) \times \oplus_{k=1}^{K} \vec{cx}_{ik}$
9　**end**
10　**for** $n \in (1 \dots N)$ **do**
11　　Initialize document vector $\vec{xd}_n = \vec{0}$
12　　**for** *word $x_i$ in $D_n$* **do**
13　　　$\vec{xd}_n = \frac{1}{|D_n|} \sum_{x \in D_n} \frac{a}{a+p(x_i)} \vec{tx}_i$
14　　**end**
15　　$\vec{xd}_{n\_pca-1} = \vec{xd}_n - PCA(First\ component)$
16　　$W\vec{SD}V_{D_n} = make\_sparse\,(\vec{xd}_{n\_pca-1})$
17　**end**

---

### 4.2. Weighted Compact Document Vector (WCDV)

The *Multimodal Word Distributions* [18] learns distributions of words using an energy-based max-margin objective. The energy function behind it is the *Expected Likelihood Kernel*, which computes the inner product between distributions to get the affinity of words. It utilizes highly expressive distributions using the Gaussian mixture model to learn multiple senses of words. However, it is unaware of maintaining the potential words in a sentence or a document. In this context, we proposed a novel energy function called *Weight attained Expected Likelihood Kernel* that considers point weights while computing the partial log energy between distributions of words. To accommodate the improvement persuing by the *Weight attained Expected Likelihood Kernel*, we redefine the objective for the *Multimodal Word Distributions* [18] and employ it to capture semantically enhanced topics for the short length document representation.

Figure 3 represents the functionality of the *Weight attained Expected Likelihood Kernel* (weighted energy function), where the kernel gets input as the Gaussian Mixture distribution of words and their corresponding point weights and produces clusters of words as the output. Figure 4 represents the overall functionality of the proposed *Weighted Compact Document Vector* (WCDV), where kernel output $\vec{cx}_k$ use to get underlying topics $\vec{tx}_i$ and finally the vectorial representation of the documents $W\vec{CD}V_{D_n}$ build using latent topics $\vec{tx}_i$.

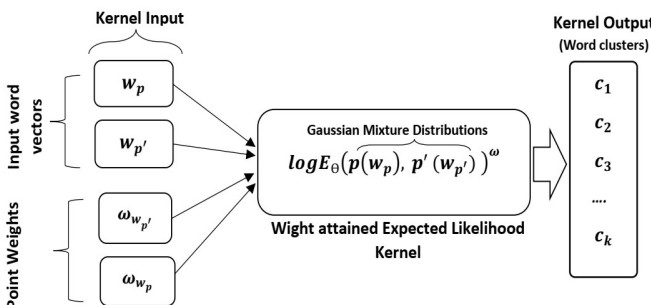

**Figure 3.** Block diagram of the semantic similarity measures between distributions of words using The Weight attained Expected Likelihood Kernel (weighted energy function).

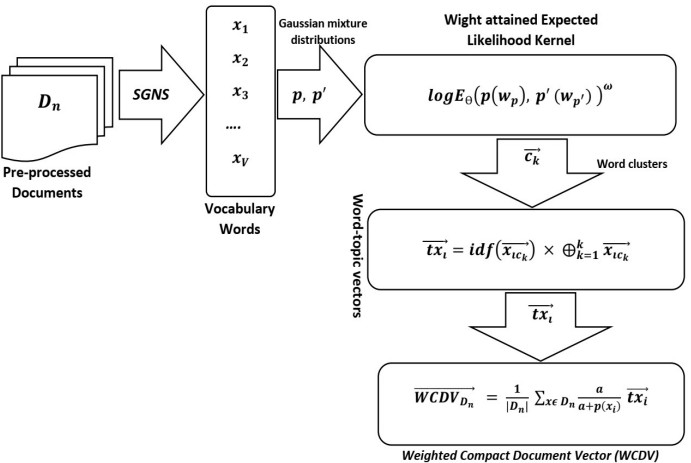

**Figure 4.** Weighted Compact Document Vector formation in to $d \times k$ dimensional embedding space.

### 4.2.1. Weight Attained Expected Likelihood Kernel

The probability product kernel $k(p, p') = \langle p, p' \rangle_{L_2}$ for $\rho > 0$ between two Gaussian distributions $p$ and $p'$ (using diagonal covariance):

$$
\begin{aligned}
\xi_{i,j}^{\rho} &\equiv \log k_{\rho}(p_i, p'_j) \\
&= (1 - 2\rho)\frac{D}{2}\log(2\pi) - \frac{D}{2}\log(\rho) \\
&+ \log det[\Sigma_{p,i}^{\rho-1} \Sigma_{p',j}^{\rho} + \Sigma_{p,i}^{\rho} \Sigma_{p',j}^{\rho-1}] \\
&- \frac{\rho}{2}(\mu_{p,i} - \mu_{p',j})(\Sigma_{p,i} + \Sigma_{p',j})^{-1}(\mu_{p,i} - \mu_{p',j})
\end{aligned}
\tag{13}
$$

where $\xi_{i,j}^{\rho}$ is partial (log) energy.

For mixture of Gaussians, the log energy is then

$$
\log E_{\theta}^{\rho}(p_i, p'_j) = \sum_{i=1}^{k}\sum_{j=1}^{k}(\pi_i \pi_j)^{\rho} e^{\xi_{i,j}}
\tag{14}
$$

when $\rho = 1$, Equation (14) is called the Expected Likelihood Kernel. The objective defined in the *Multimodal word Distributions* [18] based on Equation (14) as

$$
L_{\theta}(w, c, c') = max(0, m - \log E_{\theta}(w, c) + \log E_{\theta}(w, c'))
\tag{15}
$$

where $w$ is the current word, $c$ is the context of word $w$, $c'$ is the negative context (did not appear with the word $w$), and $m$ is an affinity margin scale.

For the Gaussian mixture distributions, two words $w_p$ and $w_{p'}$ are represented by $p_{w_p}(\vec{x}) = \Sigma_{i=1}^{k} \pi_{w_p,i} \, \mathcal{N}(\vec{x}, \vec{\mu_{w_p,i}}, \Sigma_{w_p,i})$ and $p'_{w_{p'}}(\vec{y}) = \Sigma_{j=1}^{k} \pi_{w_{p'},j} \, \mathcal{N}(\vec{y}, \vec{\mu_{w_{p'},j}}, \Sigma_{w_{p'},j})$, where, $\vec{x}$ and $\vec{y}$ are the vector representation of words $w_p$ and $w_{p'}$ respectively; mixing probabilities $\Sigma_{i=1}^{k} \pi_{w_p,i} = 1$ and $\Sigma_{j=1}^{k} \pi_{w_{p'},j} = 1$. The weights parameter $\omega$ contains relevant point weights (Equation (2)) of words $w_p$ and $w_{p'}$, which acts like the precision and has direct effects on the covariance matrix (Equation (3)). The *Weight attained Expected Likelihood Kernel* is then given by Equation (16) below:

$$
\begin{aligned}
E_\theta(p, \, p')^\omega &= \sum_{i=1}^{k} \sum_{j=1}^{k} \pi_{w_p,i} \, \pi_{w_{p'},j} \int \mathcal{N}(\vec{x}, \vec{\mu_{w_p,i}}, \frac{1}{\omega}\Sigma_{w_p,i}) . \mathcal{N}(\vec{y}, \vec{\mu_{w_{p'},j}}, \frac{1}{\omega}\Sigma_{w_{p'},j}) \, dx \\
&= \sum_{i=1}^{k} \sum_{j=1}^{k} \pi_{w_p,i} \, \pi_{w_{p'},j} \, \mathcal{N}(0; \, \vec{\mu_{w_p,i}} - \vec{\mu_{w_{p'},j}}, \, \frac{1}{\omega}\Sigma_{w_p,i} + \frac{1}{\omega}\Sigma_{w_{p'},j}) \\
&= \sum_{i=1}^{k} \sum_{j=1}^{k} \pi_{w_p,i} \, \pi_{w_{p'},j} e^{\xi_{i,j}^\omega}
\end{aligned}
\tag{16}
$$

where $\xi_{i,j}^\omega$ is the weighted partial (log) energy.

Following Equation (13), we can derive the weighted log energy for the Gaussian mixture distributions $p$ and $p'$ for words $w_p$ and $w_{p'}$ as below:

$$
\begin{aligned}
\xi_{i,j}^\omega &\equiv \log \mathcal{N}(0; \, \vec{\mu_{w_p,i}} - \vec{\mu_{w_{p'},j}}, \, \frac{1}{\omega}\Sigma_{w_p,i} + \frac{1}{\omega}\Sigma_{w_{p'},j}) \\
&= -\frac{1}{2} \log \det(\frac{1}{\omega}\Sigma_{w_p,i} + \frac{1}{\omega}\Sigma_{w_{p'},j}) - \frac{D}{2} \log(2\pi) \\
&\quad - \frac{1}{2}(\vec{\mu_{w_p,i}} - \vec{\mu_{w_{p'},j}})^T (\frac{1}{\omega}\Sigma_{w_p,i} + \frac{1}{\omega}\Sigma_{w_{p'},j})^{-1} (\vec{\mu_{w_p,i}} - \vec{\mu_{w_{p'},j}})
\end{aligned}
\tag{17}
$$

The weighted log energy is then

$$
\log E_\theta(p, \, p')^\omega = \sum_{i=1}^{k} \sum_{j=1}^{k} \pi_{w_p,i} \, \pi_{w_{p'},j} \, e^{\xi_{i,j}^\omega}
\tag{18}
$$

Equation (18) recovers Equation (16).

Based on the *Weight attained Expected Likelihood Kernel* in Equation (16), we redefine the objective (Equation (15)) for the *Multimodal word Distributions* [18] as below:

$$
L_{\theta\omega}(w, c, c', \omega) = max(0, m - \log E_\theta(w, c)^{\omega_{w,c}} + \log E_\theta(w, c')^{\omega_{w,c'}})
\tag{19}
$$

where $w$ is the current word, $c$ is the context word, $c'$ is the negative context word (which did not appear together as the context word). The weights parameter $\omega$ contains the point weight of the relevant individual words (Equation (2)).

Equation (19) defines the pair of semantically most related words with the progression of the training by pushing the similarity between the current and the context words higher than the negative context words by a margin of $m$ using the weighted log energy.

### 4.2.2. Algorithm WCDV

For the Weighted Compact Document Vector (WCDV), we employ the *Multimodal Word Distributions* [18] with the word frequency concerned energy function to compute the affinity of word pairs to capture semantically enhanced topics for the short-length documents. All the model parameters—the location (mean vector $\vec{\mu_{w_p,i}}$) of $i$th component of word $w$, covariance matrix ($\Sigma_{w_p,i}$), and the mixture weight ($\pi_{w_p,i}$) learn from the data

(corpus) using the newly defined maximum margin energy-based ranking objective using the *Weight attained Expected Likelihood Kernel*.

In Algorithm 2, line 1 is the process of computing weight (point weight) corresponding to words in the vocabulary using Equation (2). Line 4 gets clusters of wors by computing word pairs affinity using Equation (16). Line 8 gets topic embeddings $\vec{tx}_i$ by inverse document frequency (idf) weighting and cluster-wise concatenating of words. Line 13 creates *Weighted Compact Document Vector* (WCDV) by averaging topic-specific word embeddings weighted by the smooth inverse frequency ($\frac{a}{a+p(x_i)}$) [71].

---

**Algorithm 2:** WCDV

---

**Input:** Documents $D_n; n = 1 \dots N$
**Output:** Document Vectors $W C \vec{D} V_{D_n}$, in to $d \times k$ embedding space; $n = 1 \dots N$

1   Calculate *point weights* $\omega$ using Equation (2) for the corpus
2   **for** *each word paris* $(w_p, w_{p'})$ *in the vocabulary* **do**
3     **for** *each component k* **do**
4       $\vec{c}_k = \sum_{i=1}^{k} \sum_{j=1}^{k} \pi_{w_p, i} \ \pi_{w_{p'}, j} e^{\xi_{i,j}^{\omega}}$ using Equation (16)
5     **end**
6   **end**
7   **for** *each word* $\vec{x}_{ic_k}$ *in* $\vec{c}_k$ **do**
8     $\vec{tx}_i = idf(\vec{x}_{ic_k}) \times \oplus_{k=1}^{K} \vec{x}_{ic_k}$
9   **end**
10   **for** $n \in (1...N)$ **do**
11     Initialize document vector $W C \vec{D} V_{D_n} = \vec{0}$
12     **for** *word* $x_i$ *in* $D_n$ **do**
13       $W C \vec{D} V_{D_n} = \frac{1}{|D_n|} \sum_{x \in D_n} \frac{a}{a+p(x_i)} \vec{tx}_i$
14     **end**
15   **end**

---

## 5. Experimental Results and Discussion

To evaluate the proposed approaches on several publicly available datasets, we perform the experiments using Intel® Core™ i5-7500 CPU@3.40 GHz., 8GiB RAM machine with Ubuntu 16.04.7 LTS operating system.

To evaluate WSDV, we perform document classification on three publicly available datasets (long text): 20Newsgroup (almost balanced), R8 (imbalanced), and AgNews (balanced).

To evaluate WCDV, we perform document classification on two publicly available short-text datasets: SearchSnippets (balanced) and Twitter (imbalanced).

Table 1 represents the statistics of the datasets mentioned above. All datasets come with train and test subsets (by default). We use the Python NLTK (Natural language toolkit) library to remove punctuations, digits, and stopwords. We also remove extra spaces from the datasets as part of data preprocessing.

**Table 1.** Statistics of short text dataset.

| Dataset | Docs | Train | Test | Avg. Length | Vocabulary | Class |
|---|---|---|---|---|---|---|
| 20Newsgroup | 18,846 | 11,314 | 7532 | 315 | 179,209 | 20 |
| AgNews | 127,600 | 120,000 | 7600 | 39 | 72,046 | 4 |
| R8. | 7674 | 5485 | 2189 | 64 | 16,698 | 8 |
| SearchSnippets | 12,265 | 10,050 | 2215 | 10.7 | 5581 | 8 |
| Twitter | 5113 | 3513 | 1600 | 5.0 | 1390 | 4 |

For the classification task, We use a 5-fold cross-validation on the Accuracy and F1 score to tune parameter "C" of the Support Vector Classifier. Using preprocessed datasets, we use the train portion to fit the classifier and the test portion to evaluate the results. We employ Scikitlearn GridSearchCV (https://scikit-learn.org/stable/modules/generated/sklearn.model_selection.GridSearchCV.html (accessed on 10 June 2022)) with sklearn.svm.SVC (https://scikit-learn.org/stable/modules/generated/sklearn.svm.SVC.html (accessed on 10 June 2022)) Support Vector Classification for model fitting and performing classification. In $GridSearchCV$, we set $LinearSVC(c = 1)$, $param\_grid = ['C' : np.range(0.1, 7, 0.2)]$, $cv = 5$, $n\_jobs = 5$ as parameters.

### 5.1. Document Classification (Long Text)

We compare the classification results of the WSDV with a study was done by Wagh, V. et al. [73], as their experiments range from simple NaiveBayes to complex BERT approaches intending to compare the classification performance of machine learning algorithms under the same set of long document datasets. To avoid training complexity (e.g., computational environment, parameter settings of different models) of the baselines, we prefer to use the obtained results by Wagh, V. et al. [73] as the standard for all three (long text) datasets mentioned above.

To train WSDV, we set $\gamma$ = 7 in Equation (5). For other hyper parameters settings, we follow similar settings as SCDV [6], such as word embedding dimension to 200, document vector sparsity threshold to 0.04, minimum word count to 20, window size to 10, and downsampling to $10^{-3}$.

For the classification, we set the number of topics to 20 (ground truth) for 20Newsgroup, 8 (ground truth) for R8, and 4 (ground truth) for AgNews. Table 2 illustrates classification accuracy compared with the proposed WSDV and the baselines [73] using different datasets (long-text), where bold indicates the best results in the table. From Table 2 we see, WSDV obtaines superior scores in terms of accuracy—about 97.83% accuracy using AgNews, about 86.05% accuracy using 20Newsgrup and about 98.76% accuracy using R8 datasets.

**Table 2.** Classification accuracy (%) evaluation using different datasets.

| Model | AgNews | 20Newsgroup | R8 |
| --- | --- | --- | --- |
| TFIDF with Naive-Bayes | 90.45 | 81.69 | 84.24 |
| GloVe+Average | 92.07 | 80.43 | 95.57 |
| GloVe+Attention | 92.39 | 81.65 | 95.61 |
| LSTM+CNN | 92.71 | 79.74 | 97.17 |
| BiLSTM+Max | 92.59 | 83.02 | 97.03 |
| BiLSTM+Attention | 93.14 | 81.76 | 95.80 |
| USE | 92.09 | 81.76 | 95.61 |
| ULMFiT | 94.00 | 82.4 | 96.48 |
| HAN | 92.11 | 85.01 | 94.47 |
| BERT | 94.04 | 85.78 | 97.62 |
| DistilBERT | 94.02 | 85.43 | 97.53 |
| WSDV | **97.83** | **86.05** | **98.67** |

WSDV is closely related to SCDV and SCDV-MS. For the efficiency assessment of the proposed WSDV, we compare the time and space complexity, sparsity analysis, and the obtained F1 scores using the 20Newsgroup corpus. For the fair assessment, we use SCDV-MS with word2vec instead of the Doc2vecC version in the comparison, as both other (WSDV and SCDV) models use SkipGram negative sampling (SGNS). We use an unlabeled 20Newsgroup corpus and set the number of topics to 20. Table 3 represents class-wise F1 scores obtained by WSDV, SCDV, and SCDV-MS (word2vec), where bold indicates the best results in the table. For the experiments in this section, we use the default parameters settings for SCDV and SCDV-MS.

**Table 3.** Class-wise F1-Score comparison among the three most related models using the 20Newsgroup dataset (20 clusters).

| Class Name | WSDV | SCDV | SCDV-MS (word2vec) |
|---|---|---|---|
| alt.atheism | 0.819355 | 0.789727 | **0.821830** |
| comp.graphics | 0.750000 | 0.738035 | **0.752525** |
| comp.os.ms-windows.misc | **0.754098** | 0.739300 | 0.727742 |
| comp.sys.ibm.pc.hardware | 0.712846 | **0.724005** | 0.684796 |
| comp.sys.mac.hardware | 0.812261 | **0.832250** | 0.784211 |
| comp.windows.x | 0.808625 | **0.812332** | 0.787798 |
| misc.forsale | **0.844784** | 0.834805 | 0.766709 |
| rec.autos | 0.898734 | **0.906566** | 0.867008 |
| rec.motorcycles | **0.940431** | 0.931373 | 0.928136 |
| rec.sport.baseball | **0.946716** | 0.930991 | 0.936709 |
| rec.sport.hockey | **0.977612** | 0.967980 | 0.970149 |
| sci.crypt | **0.928040** | 0.927681 | 0.912060 |
| sci.electronics | **0.751309** | 0.738186 | 0.741772 |
| sci.med | **0.886275** | 0.886010 | 0.861538 |
| sci.space | **0.923077** | 0.911111 | 0.877680 |
| soc.religion.christian | 0.868852 | **0.876325** | 0.858156 |
| talk.politics.guns | **0.815920** | 0.818955 | 0.812579 |
| talk.politics.mideast | 0.942779 | **0.947083** | 0.952255 |
| talk.politics.misc | 0.695035 | 0.671587 | **0.714286** |
| talk.religion.misc | **0.647450** | 0.616740 | 0.630872 |

Table 4 illustrates an empirical study of the time and space complexities of WSDV, SCDV, and SCDV-MS (word2vec). WSDV deals with weighted data clustering, where every data point is associated with a unique point weight. Weighted data clustering takes a little extra time for the clustering process of WSDV (22.36 s). However, it carries faster (0.54 s) prediction characteristics inherited from the *Pomegranate General Mixture* model [16] that acquires less memory space (66.9 kb). When comparing WSDV (multi-sense) with SCDV-MS (word2vec), again WSDV (multi-sense) takes a higher clustering time (104.6 s) but faster (0.96 s) prediction time than SCDV-MS (word2vec) and takes less memory space (102.1 kb).

**Table 4.** Time and Space complexity analysis using the 20Newsgroup dataset (20 clusters).

| Model | Vocab | $\vec{wtv}$ Dim | Cluster (s) | Prediction (s) | Model (kb) |
|---|---|---|---|---|---|
| WSDV | 16,676 | 4000 | 22.36 | **0.54** | **66.9** |
| SCDV | 16,676 | 4000 | **19.52** | 0.71 | 133.6 |
| WSDV (multi-sense) | 25,465 | 4000 | 104.6 | <u>0.96</u> | <u>102.1</u> |
| SCDV-MS (word2vec) | 25,465 | 4000 | 40.19 | 1.20 | 203.9 |

Table 5 demonstrates sparsity level in the word-topic $\vec{wtv}$ vector and document $\vec{wtv}$ vector. WSDV applies a threshold to remove outliers (similar to SCDV-MS), which leads to higher sparsity (94.85%) in the word-topic $\vec{wtv}$ vector. When comparing WSDV (multi-sense) with SCDV-MS (word2vec), WSDV has an additional document vector sparsity mechanism (similar to SCDV) and achieves higher (65.50%) sparsity in the final document representation than SCDV-MS (word2vec).

**Table 5.** Document Sparsity analysis using the 20Newsgroup dataset (20 clusters).

| Model | Vocab | $\vec{wtv}$ Dim | $\vec{wtv}$ Sparsity (%) | $\vec{dv}$ Sparsity (%) |
|---|---|---|---|---|
| WSDV | 16676 | 4000 | 94.85 | 63.54 |
| SCDV | 16676 | 4000 | (0.01) n/a | **66.19** |
| WSDV (multi-sense) | 25465 | 4000 | **95.04** | 65.50 |
| SCDV-MS (word2vec) | 25465 | 4000 | **95.04** | 27.16 |

Table 6 exhibits the effects of weighted data clustering and re-ranking the top words for topical representation of WSDV over SCDV. *Topic*1 related to sports where WSDV represents a better topic (with the PMI score −372.469) than SCDV (with the PMI score −373.966). After re-ranking the top words, WSDV improves topic quality with a PMI score of −181.043. *Topic*2 related to IT, it is interesting to see that both the SCDV (PMI score −403.195) and the WSDV (PMI score −404.763) represent similar top 10 words. After re-ranking, *WSDV* surpasses both scores by obtaining an improved PMI score of −230.419. *Topic*3 related to vehicles, though *WSDV* shows a better PMI score (−380.499) than the SCDV (−394.077), SCDV seems to represent more relevant words than WSDV. Again after re-ranking, WSDV shows more relevant words (with the PMI score −210.754). Table 6 also proves the necessity of clustering on weighted data for exuberant document representation.

**Table 6.** Top words and Coherence scores of few topics using 20newsgroup dataset (20 clusters).

| | Topic 1 | | | Topic 2 | | | Topic 3 | |
|---|---|---|---|---|---|---|---|---|
| SCDV | WSDV | WSDV (Re-Rank) | SCDV | WSDV | WSDV (Re-Rank) | SCDV | WSDV | WSDV (Re-Rank) |
| Game | Game | Game | Nx | System | Windows | Power | Car | Car |
| Team | John | Team | System | Windows | File | Car | Heard | Bike |
| Win | Team | Hockey | Windows | File | Drive | Buy | Couple | Cars |
| Games | St | Players | File | Software | Dos | Price | Bike | Engine |
| Play | Win | Season | Software | Bit | Card | Sale | Mine | Miles |
| Hockey | Games | Nhl | Bit | Number | Image | Light | Cars | Bmw |
| Canada | Play | Teams | Data | Data | Files | Bike | Recently | Ride |
| St | Vs | Leafs | Drive | Drive | Pc | Speed | Gas | Battery |
| Division | Toronto | Bruins | Program | Program | Graphics | Cost | Insurance | Radar |
| Red | Hockey | Scoring | Version | Version | Mac | Model | Engine | Riding |
| −373.966 | −372.469 | **−181.043** | −403.195 | −404.763 | **−230.419** | -394.077 | −380.499 | **−210.754** |

From the discussion above, we see the efficiency of the WSDV in the multi-class document classification tasks over SCDV and other baselines.

*5.2. Short Text Classification*

Yi, F. et al. proposed a regularized non-negative matrix factorization topic model (TRNMF) [12] for short text. TRNMF utilized pre-trained distributional vector representation of words using an external corpus and employed a clustering mechanism under document-to-topic distribution. One of our research objectives was to model efficient sparse short-length documents utilizing available corpus statistics instead of depending on external knowledge sources. Therefore, for a fair assessment of the classification performance of the proposed WCDV using the same (SearchSnippet and Twitter) datasets and to avoid train complexity of the baselines, we consider the obtained short text classification scores in the study done by Yi, F. et al. [12] as standard for the baselines.

To train WCDV, we follow the same parameters settings as *Multimodal word Distributions* [18], such as embedding size to 50, window size to 5, batch size to 256, train epoch to 10, variance scale to 0.05, and choose spherical covariance matrix. To compare short text classification accuracy with baselines, in WCDV, we set the number of topics



to 8 (ground truth) and 4 (ground truth) for the SearchSnippets and the Twitter dataset correspondingly. Table 7 illustrates classification accuracy obtained by baselines [12] and the proposed WCDV using SearchSnippets and Twitter datasets, where WCDV shows superior scores, about 72.7% accuracy using SearchSnippets, and about 89.4% accuracy using Twitter datasets.

**Table 7.** Classification accuracy (%) evaluation using Short text datasets.

| Model | SearchSnippets | Twitter |
|---|---|---|
| BTM | 19.8 | 40.9 |
| WNTM | 46.8 | 77.6 |
| LF-DMM | 17.1 | 69.5 |
| GPU-DMM | 15.1 | 48.7 |
| PTM | 10.7 | 43.1 |
| TRNMF | 56.9 | 79.9 |
| WCDV | **72.7** | **89.4** |

Short texts tend to have a small number of themes. Besides performing the classification accuracy under the ground truth (Table 7), similar to Yi, F. et al. [12], we also perform short text classification under the number of topics to 10 (ten) for both (SearchSnippets and Twitter) datasets to explore F1 scores obtained by the proposed WCDV. The F1 score is a statistical explanation of the classification performance; higher F1 scores indicate better classification performance. Figure 5 exhibits the obtain F1 scores. TRNMF has the best (55.1%) F1 score compared to other benchmark methods (BTM 24.2%, WNTM 50.8%, LF-DMM 18.9%, GPU-DMM 34.3%, PTM 16.6%) using the SearchSnippet dataset, where WCDV surpass all the baselines (including TRNMF) by obtaining 60.7% F1 score.

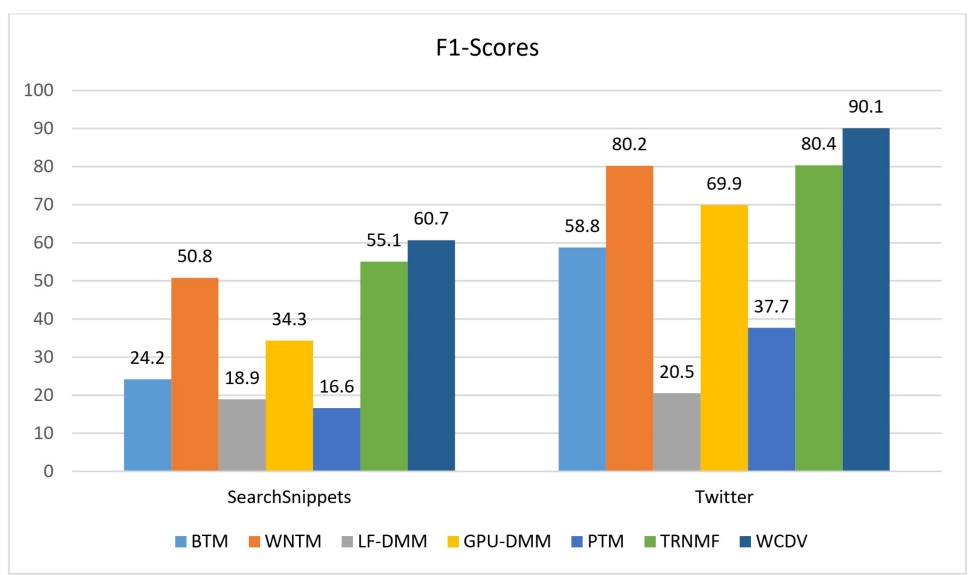

**Figure 5.** Exploring F1 Scores for 10 ($k = 10$) topics using SearchSnippets and Twitter datasets.

Similarly, WCDV shows outstanding performance over the baselines [12] using the Twitter dataset, where WCDV obtains the highest F1 score of 90.1 %, which surpass the best baseline model TRNMF (80.4%) by the margin of 9.7% improvement.

For all classification experiments using WCDV, we set the minimum word count to 10 for the SearchSnipts dataset and 5 for the Twitter dataset. The *Multimodal word Distributions* [18] has proved effective in the polysemous word representation. To evaluate the efficiency of the newly proposed objective (Equation (19)), we compare nearest neighbors (cosine similarity) between the Gaussian mixtures (components $k = 2$) mean vectors for the base $L_\theta$ (Equation (15)) and the new $L_{\theta\omega}$ (Equation (19)) objectives.

For this experiment, we use the Text8 dataset and set the number of topics to 2, the embedding dimensions to 50, the context window size to 10, the learning rate to 0.05, batch size to 128, the number of epochs to 10, use *adagard* as the optimizer, and choose *spherical* covariance model with variance scale 0.05. For both objectives, we use Tensorflow 1.5.0.

Table 8 contains three words corresponding to two components (namely 0 and 1) of the mixtures of Gaussian. The notation $w:i$ denotes the $i$th mixture component of the word $w$.

**Table 8.** Nearest neighbor words (cosine similarity) using text8 dataset for both objectives ($L_\theta$ and $L_{\theta\omega}$) in the *Multimodal word Distributions* settings.

| Rock0 | | Rock1 | | Bank0 | |
|---|---|---|---|---|---|
| $L_\theta$ | $L_{\theta\omega}$ | $L_\theta$ | $L_{\theta\omega}$ | $L_\theta$ | $L_{\theta\omega}$ |
| band:1 | albums:0 | rocks:1 | crust:0 | airport:1 | corner:0 |
| rap:1 | jazz:0 | uplifted:0 | sand:0 | kahului:0 | islet:0 |
| musicians:0 | song:0 | limestone:0 | rolling:1 | eilat:1 | chester:1 |
| music:1 | devo:1 | stalactites:0 | rocky:0 | mostar:1 | northwards:0 |
| punk:0 | songwriting:1 | sand:1 | volcanic:0 | omari:1 | holyhead:1 |
| pop:1 | rap:1 | meltwater:1 | undulating:1 | skyscraper:0 | evansville:1 |
| reggae:1 | album:0 | dunes:1 | thick:0 | caucasoid:0 | dobruja:1 |
| blues:1 | funk:1 | sandy:0 | outcrops:1 | dubai:1 | sacramento:1 |
| disco:0 | grateful:1 | melts:1 | boulders:0 | kyiv:1 | shambles:0 |
| Bank1 | | Apple0 | | Apple1 | |
| $L_\theta$ | $L_{\theta\omega}$ | $L_\theta$ | $L_{\theta\omega}$ | $L_\theta$ | $L_{\theta\omega}$ |
| monetary:1 | banking:0 | mac:1 | macintosh:0 | ibm:1 | mint:0 |
| eurozone:1 | monetary:0 | marketed:0 | microsoft:0 | macintosh:0 | brandy:1 |
| banking:0 | investment:0 | desktop:1 | ibm:1 | microsoft:1 | apples:1 |
| eu:0 | privatization:1 | fermented:0 | amiga:0 | amiga:0 | fried:1 |
| businesses:1 | imf:1 | sourced:0 | desktop:0 | beos:0 | liqueur:0 |
| sector:1 | multilateral:0 | oyster:0 | console:1 | windows:1 | quince:0 |
| exchange:0 | revenue:1 | micro:0 | intel:0 | developers:0 | fruit:0 |
| loans:0 | exporting:0 | portable:0 | powerpc:1 | hardware:1 | juice:1 |
| currencies:1 | billion:1 | pies:0 | macromedia:1 | microprocessor:0 | juicy:1 |

Words Rock and Bank represent the right theme corresponding to each mixture component for both objectives. However, the new objective $L_{\theta\omega}$ works in more detail than the base objective $L_\theta$. Bank0 represents a specific area of a place. Words for $L_{\theta\omega}$ in the table pose this characteristic, which is not true for words represented by $L_\theta$, specifically, in $L_\theta$ word mostar:1 is the name of a person, which is irrelevant. Bank1 represents a theme related to finance, where for $L_{\theta\omega}$ words are relevant. For $L_\theta$ eu:1, sector:1 and eurozone:1 do not directly hold financial sense. However, for the word Apple we found inconsistency for the base objective $L_\theta$, where Apple1 and Apple0 represent the same theme. In the case of Apple0, the nearest neighbor words pies:0, oyster:0, and fermented:0 are neither related to the computer technology nor the fruit; those may relate to the food theme, but other neighboring words represent the computer-related sense. However, the new objective $L_{\theta\omega}$ represents the fruit (particularly apple; food, and drinks processed from apple; quince:0 is a fruit related to apple.) theme for Apple1 and the computer technology theme for Apple0 properly.

From the overall analysis, we see the new objective $L_{\theta\omega}$ performs better than the base objective $L_\theta$ in terms of the similarity measure of the polysemous words. It proves the efficiency of the *Weight attained Expected Likelihood Kernel* as a novel energy function.

### 5.3. Discussion

This research focused on utilizing available corpus statistics to enhance document classification performance, where we proposed *Weighted Sparse Document Vector* (WSDV) for the long text and *Weighted Compact Document Vector* (WCDV) for the short text.

SCDV [6] and SCDV-MS [13] have achieved enormous performance by addressing the challenges in the document (long text) modeling. However, we proposed further escalation in the document classification by introducing WSDV. Designing WSDV includes emphasizing potential terms of the document in the corpus (using weighted data clustering), noise elimination (soft threshold noise removal) from the noisy long tail clusters, and utilizing corpus statistics in different steps of the vectorial representation of documents. Experiments in Section 5.1 demonstrate that WSDV significantly outperforms SCDV and SCDV-MS in document modeling (long text). Moreover, WSDV efficiently handles balanced (AgNews) and imbalanced (R8) corpus for the classification task (according to Table 2). Therefore, WSDV is an excellent addition to the existing state-of-the-art long text classification approaches.

Most state-of-the-art short text models suggest acquiring knowledge from external rich knowledge sources to tackle data sparseness. However, depending on the external knowledge sources is not reliable (e.g., data unavailability, missing URLs, etc.). Sometimes it may increase unusual noise in the current corpus or cause higher costs. Therefore, we proposed *Weighted Compact Document Vector* (WCDV). WCDV utilizes corpus statistics in different steps of the vectorial representation of short-length documents. Experiments reveal that WCDV efficiently deals with sparse short texts without depending on external knowledge sources with balanced (SearchSnippets) and imbalanced (Twitter) datasets (according to Table 7, and Figure 5). Moreover, we have introduced a novel energy function to capture the affinity of the distributions, which emphasizes the potential terms by assigning corresponding point weights to them. Experiment in Section 5.2 demonstrate that the proposed *Weight attained Expected Likelihood Kernel* is an excellent addition to the similarity kernel and performs better than its counterpart.

## 6. Conclusions

Overall experiments witness the undoubted enhancement of the document representation capability by the proposed WSDV and WCDV that utilize corpus statistics to improve document classification for noisy long and sparse short texts.

WSDV represents an expressive document by logically dividing the embedding space into the number of clusters semantic subspaces, which involves clustering on weighted data (using *point weights* in the clustering process) and utilizes corpus statistics at different levels of document representation. Experiments demonstrate that along with reducing noise on the weight data clustering, using *inverse document frequency* (IDF) for topic construction and *smooth inverse frequency* (SIF) in the document embedding made WSDV robust for the long texts classification. WCDV successfully operates on the sparse short text by capturing better semantic insights using proposed *Weight attained Expected Likelihood Kernel* that emphasizes potential terms. Experiments demonstrate that without acquiring external knowledge, the newly proposed energy function alongside *inverse document frequency* (IDF) for topic construction and *smooth inverse frequency* (SIF) in the document embedding sufficiently performs for short-length documents classification.

Moreover, this study will positively impact future Natual Language Processing (NLP) research. The proposed *Weight attained Expected Likelihood Kernel* has opened a new door in distribution level similarity measuring for future downstream tasks (e.g., image processing). On the other hand, the Weighted data clustering combined with the noise elimination technique may utilize in different downstream Natural Language Processing (NLP) studies for the desired performance.

**Author Contributions:** Conceptualization, formal analysis, investigation, methodology, and writing—original draft, F.U.; Supervision, Z.Z.; review and validation, Z.Z., Y.C. and X.H.; funding acquisition, Z.Z., Y.C. and X.H. All authors have read and agreed to the published version of the manuscript.

**Funding:** This research was funded by Hunan Key Laboratory for Internet of Things in Electricity (Grant No. 2019TP1016) and the National Natural Science Foundation of China (Grant

No.72061147004) the National Natural Science Foundation of Hunan Province (Grant No. 2021JJ30055) and the project about research on key technologies of power knowledge graph (Grant No. 5216A6200037).

**Data Availability Statement:** All datasets used for the evaluation of the proposed approaches in this article are publicly accessible through the following URLs: 20Newsgroup: http://qwone.com/~jason/20Newsgroups/ (accessed on 10 June 2022), SearchSnippets (Snippets): https://www.kaggle.com/mantunes/semantic-corpus-from-web-search-snippets (accessed on 10 June 2022), AGNews: http://www.di.unipi.it/~gulli/AG_corpus_of_news_articles.html (accessed on 10 June 2022), Router (R8): https://www.kaggle.com/datasets/weipengfei/ohr8r52 (accessed on 10 June 2022), Twitter: https://github.com/zfz/twitter_corpus (accessed on 10 June 2022), Text8: https://deepai.org/dataset/text8 (accessed on 10 June 2022).

**Conflicts of Interest:** The authors declare that they have no conflict of interest with respect to the research, authorship, and/or publication of this article.

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
