# Peer review of "Corpus Statistics Empowered Document Classification"

_electronics, doi:10.3390/electronics11142168_

Round 1

Reviewer 1 Report

Important study has been done on document classification and a Weighted Compact Document Vector (WCDV) has been proposed. I think that the study will make important contributions to the literature. However, there are some shortcomings that I have identified in the study. These are listed below. Please review these suggestions carefully.

1) Numerical information about the study should be included in the abstract section. What is the gain with the proposed approach? It should be explained.

2) What makes the proposed method innovative? What will be its contribution to the literature? How is it different from other methods? These questions should be answered in the Introduction Section.

3) What are the research questions of this study? The Introduction Section should develop in this direction.

4) Figure 3 and Figure 4 should be given larger. In addition, Figure 3 and Figure 4 should be explained in detail in the text.

5) The proposed approach is compared with different methods in Table 1 and Table 6. How the results of the methods given here were obtained. Why were these methods chosen for comparison?

6) Figure 5 should be explained in detail.

7) The discussion of the study is insufficient.  A separate Discussion Section should be created for a better understanding of the study.

8) How will this study guide other future studies? It should be given in the Conclusion or Discussion Section.

Author Response

Q#1. Numerical information about the study should be included in the abstract section. What is the gain with the proposed approach? It should be explained.

Answer:

We have modified the “Abstract”, which contains the purpose of conducting this study and the proposed solution regarding the current shortcomings.

Finally, we have mentioned the gain of the proposed approaches with numerical information in lines 15-21:

“To evaluate proposed models, we perform multiclass document classification using standard performance measures (precision, recall, f1-score, and accuracy) on three long and two short text benchmark datasets that outperform some state-of-the-art models. Experimental results demonstrate that in the long texts classification, WSDV reached 97.83 % accuracy on the AgNews dataset, 86.05 % accuracy on the 20Newsgroup dataset, and 98.67 % accuracy on the R8 dataset. In the short texts classification, WCDV reached 72.7 % accuracy on the SearchSnippets dataset and 89.4 % accuracy on the Twitter dataset. ”

Q#2. What makes the proposed method innovative? What will be its contribution to the literature? How is it different from other methods? These questions should be answered in the Introduction Section.

Answer:

We have followed these questions throughout writing the Introduction Section. Here are some important statements.

We have stated the problem with the current long text classification methods (Specially SCDV) in lines 40-44:

“However, SCDV inherits noisy tails from the Gaussian mixture clustering that is not appropriate for the document containing multiple sentences. Another important issue ignored by most document representation methods is ignoring potential terms in the corpus, which is vital for understanding deep semantic insight of the documents.”

 , and the problem with its improved version (SCDV-MS) in lines 60-65:

“Sparse Composite Document Vector with Multi-Sense Embeddings (SCDV-MS) forces discarding the outliers from the clustering output to eliminate long-tail noises in the SCDV, which applies a hard threshold that may hinder the thematic representation of documents. Moreover, representing proper expressive documents depends upon modeling the underlying semantic topics in the correct form, which requires capturing deep semantics insights buried in words, expressions, and string patterns.”

We have stated our solution for these in lines 65-69:

“Hence, for the noisy long texts, we propose Weighted Sparse Document Vector (WSDV), which consists important words emphasizing capability using an alternative to the Gaussian mixture model called Pomegranate General Mixture model and a soft threshold-based noise reduction technique.”

We have stated the problem with the current short text models in lines 45-49:

“It is hard to encode the richness of semantic insights for short-length documents where word co-occurrence information is limited. Therefore, many works suggest importing semantic information from external sources. However, accessing information from external sources (e.g., Wikipedia) may cause irrelevant noise corresponding to the current short text corpus.”

We have stated our solutions for the short text classification in lines 70-85:

“It is challenging to capture semantics insights in document modeling with sparse short texts. The probability distribution of words captures better semantics than the point embedding approach (e.g., word2vec) as it generalizes deterministic point embeddings of the terms using the mean vector, and the covariance matrix holds uncertainty of the estimations. Hence, instead of depending on external knowledge sources, we proposed corpus statistics empowered Weighted Compact Document Vector (WCDV), which emphasizes potential terms while performing probability word distribution using Gaussian mixture model. In WCDV, we employ the Multimodal word Distributions that learns distributions of words using the Expected Likelihood Kernel, which computes the inner product between distributions of words to get the affinity of word pairs. However, every word in a document does not hold the same importance; some are used more frequently than others, indicating their importance in the corpus. It is required to emphasize the frequently used words, especially when word co-occurrence information is limited (e.g., microblogging, product review, etc.). Therefore, to preserve the word frequency importance, we propose Weight attained Expected Likelihood Kernel which considers term frequency-based point weights while measuring the partial log energy between distributions in the Multimodal word Distributions.”

Q#3. What are the research questions of this study? The Introduction Section should develop in this direction.

Answer:

In writing introduction section, we have reviewed some popular existing document modeling methods with their advantages and drawbacks then we have stated the research objectives as a set of research questions to solve. (lines 45-59)

“In this context, we aim to develop corpus statistics-based semantically enriched vectorial representation of the noisy long and sparse short texts for the multi-class document classification performance with the objectives of solving the following research questions:

1) How to efficiently model noisy long and sparse short texts for the classification performance?

2) How to efficiently encode the semantically enriched vectorial representation of documents using available corpus statistics to enhance classification performance?

3) Is it possible to model efficient sparse short-length documents utilizing available corpus statistics instead of depending on external knowledge sources?

 4) How can we utilize potential words in the document for deep thematic insights?”

Finally, we have provided the solution, innovations, etc., which related to the previous question (Q#2).

Q#4. Figure 3 and Figure 4 should be given larger. In addition, Figure 3 and Figure 4 should be explained in detail in the text.

Answer:

We have given expanded (modified) and larger figures (Figure 3 and Figure 4) and explanations of their functionalities in lines 357-362.

Q#5. The proposed approach is compared with different methods in Table 1 and Table 6. How the results of the methods given here were obtained. Why were these methods chosen for comparison?

Answer:

For Table 1 (Now Table 2) we have explained it in lines 421-427.

“We compare the classification results of the WSDV with a study was done by Wagh, V. et al. as their experiments range from simple NaiveBayes to complex BERT approaches intending to compare the classification performance of machine learning algorithms under the same set of long document datasets. To avoid training complexity (e.g., computational environment, parameter settings of different models) of the baselines, we prefer to use the obtained results by Wagh, V. et al. as the standard for all three (long text) datasets mentioned above.”

For Table 6 (now Table 7) we have explained in lines 479-487.

“Yi, F. et al. proposed a regularized non-negative matrix factorization topic model (TRNMF)  for short text. TRNMF utilized pre-trained distributional vector representation of words using an external corpus and employed a clustering mechanism under document-to-topic distribution. One of our research objectives was to model efficient sparse short-length documents utilizing available corpus statistics instead of depending on external knowledge sources. Therefore, for a fair assessment of the classification performance of the proposed WCDV using the same (SearchSnippet and Twitter) datasets and to avoid train complexity of the baselines, we consider the obtained short text classification scores in the study done by Yi, F. et al. as standard for the baselines.”

Q#6. Figure 5 should be explained in detail.

Answer:

We have given a detail explanation of Figure 5 in lines 496-507.

“Short texts tend to have a small number of themes. Besides performing the classification accuracy under the ground truth (Table 7), similar to Yi, F. et al., we also perform short text classification under the number of topics to 10 (ten) for both (SearchSnippets and Twitter) datasets to explore F1 scores obtained by the proposed WCDV. The F1 score is a statistical explanation of the classification performance; higher F1 scores indicate better classification performance. Figure 5 exhibits the obtained F1 socres, where TRNMF has the best (55.1 %) F1 score compared to other benchmark methods (BTM 24.2 %, WNTM 50.8 %, LF-DMM 18.9 %, GPU-DMM 34.3 %, PTM 16.6 %) using the SearchSnippet dataset, where WCDV surpass all the baselines (including TRNMF) by obtaining 60.7 % F1 score. Similarly, WCDV shows outstanding performance over the baselines using the Twitter dataset, where WCDV reached the highest F1 score of 90.1 %, which surpasses the best baseline model TRNMF (80.4 %) by the margin of 9.7 % improvement.”

Q#7. The discussion of the study is insufficient. A separate Discussion Section should be created for a better understanding of the study.

Answer:

We have created a separate Discussion subsection in lines 540-567.

Q#8 How will this study guide other future studies? It should be given in the Conclusion or Discussion Section.

Answer:

We have modified the Conclusion Section. Lines 584-589 stated a brief guidance for the future studies.

“Moreover, this study will positively impact future Natual Language Processing (NLP) research. The proposed Weight attained Expected Likelihood Kernel has opened a new door in distribution level similarity measuring for future downstream tasks (e.g., image processing). On the other hand, the Weighted data clustering combined with the noise elimination technique may utilize in different downstream Natural Language Processing (NLP) studies for the desired performance.”

Reviewer 2 Report

The idea in this paper titled "Corpus Statistics Empowered Document Classification" is good. The overall paper is very well written. However, the authors are suggested to address the following comments while revising the paper.

1: Very limited number of recent 2021, and no 2022 papers are referred to in this manuscript. It is suggested to extend the related work section by adding manuscripts from 2021, and 2022.
2: Line 229: Weighted Sparse Document Vectors (WSDV) (Figure 2). Verify the figure number, as it starts from figure 2 in the text.
3: Line 232: (Figure 4), also verify it and review the overall paper carefully for typos and other mistakes.
4: A number of language models are mentioned in Table 1 such as BERT. However, they are not introduced before in literature or any other section. The authors are suggested to review and add recent studies on these models as well such as for BERT "Effectiveness of Fine-Tuned BERT Model in Classification of Helpful and Unhelpful Online Customer Reviews. Electronic Commerce Research (2022): 1-21.".
5: Hyperparameter tuning is performed or not for models given in Table 1? Provide the details of hyperparameters used as well.
6: Provide the descriptive statistics of datasets used in a more detailed manner and in a tabular format. Also clearly state how data is split into train and test datasets. What was the split ratio? The dataset is balanced or not? etc.

Author Response

Q#1. Very limited number of recent 2021, and no 2022 papers are referred to in this manuscript. It is suggested to extend the related work section by adding manuscripts from 2021, and 2022.

Answer:

We have extended the related work section (Literature Review), where References 21, 36, 37, and 47 are from the year 2021. References 23, 25, 28, 29, and 39 are from 2022, and References 24, 26, 30, 40, 41, 46, and 48 are from the year 2020.

Q#2. Line 229: Weighted Sparse Document Vectors (WSDV) (Figure 2). Verify the figure number, as it starts from figure 2 in the text.

Answer:

We have verified Figure 1 (line 310) and Figure 2 (line 312).

Q#3. Line 232: (Figure 4), also verify it and review the overall paper carefully for typos and other mistakes

We have verified Figure 3 (line 357) and Figure 4 (line 359). We have made effort to review the article for typos and other mistake, such as remove equation 19 and equation 20 at line 318.

Q#4. A number of language models are mentioned in Table 1 such as BERT. However, they are not introduced before in literature or any other section. The authors are suggested to review and add recent studies on these models as well such as for BERT "Effectiveness of Fine-Tuned BERT Model in Classification of Helpful and
Unhelpful Online Customer Reviews. Electronic Commerce Research (2022): 1-21."

Answer:

We have modified the Introduction and the Literature Review Sections, where we have given recent works references / discussion about the models mentioned in Table 1 (Now it is Table 2) and Table 7. For example, References 2, 3 (at line 29), 21 (at line 96), 22  (at line 99), 23 (at line 102), 27 (at line 101), 28 (at line 113), 29 (at line 115), 12 (at line 47), etc. "Effectiveness of Fine-Tuned BERT Model in Classification of Helpful and
Unhelpful Online Customer Reviews. Electronic Commerce Research (2022): 1-21." is the reference number 25 at line 105.

Q#5. Hyperparameter tuning is performed or not for models given in Table 1? Provide the details of hyperparameters used as well.

Answer:

We compare the classification results of the WSDV with a study was done by Wagh, V. et al. [73], as their experiments range from simple NaiveBayes to complex BERT approaches intending to compare the classification performance of machine learning algorithms under the same set of long document datasets. To avoid training complexity (e.g., computational environment, parameter settings of different models) of the baselines, we prefer to use the obtained results by Wagh, V. et al. [73] as the standard for all three (long text) datasets mentioned above. (Please see line 421-427).

For WSDV, we use the same hyperparameter settings as SCDV [6], therefore we did not do any hyperparameter tuning. Major hyperparameters settings are mentioned in line 428-431.

Q#6. Provide the descriptive statistics of datasets used in a more detailed manner and in a tabular format. Also clearly state how data is split into train and test datasets. What was the split ratio? The dataset is balanced or not? etc.

Answer:

We have provided descripted statistics in Table 1.  We have collected split train and test subsets, Table 1 shows their train and test corpus ratio. All datasets used in this research for the evaluation of the proposed approaches are publicly accessible through the given URLs in the “Data Availability Statement”. Datasets used in this research were both balanced and imbalanced. To evaluate WSDV, we perform document classification on three publicly available datasets (long text): 20Newsgroup (almost balanced), R8 (imbalanced), and AgNews (balanced). [line 403-404].

To evaluate WCDV, we perform document classification on two publicly available short-text datasets: SearchSnippets (balanced) and Twitter (imbalanced). [line 405-406]

Round 2

Reviewer 1 Report

The authors have made the suggested updates. The study is acceptable as it is.